# Numerical Modeling Rolling Contact Problem and Elasticity Deformation of Rolling Die under Hot Milling

**Mesay Alemu Tolcha** [1,*,†] **and Holm Altenbach** [2,*,†]

1    Jimma Institute of Technology, Jimma University, Jimma 378, Ethiopia
2    Fakultät für Maschinenbau, Otto-von-Guericke-Universität Magdeburg, 39106 Magdeburg, Germany
\*    Correspondence: alemu170@yahoo.com (M.A.T.); mesay.tolcha@ovgu.de (H.A.)
†    These authors contributed equally to this work.

**Abstract:** In metalworking, rolling is a metal-forming process in which slab is passed through one or more pairs of the rolling dies to reduce the thickness and to make the thickness uniform. Modeling of rolling die contact with the slab primarily needs to describe the Tribology of contact phenomena. The central concern of numerical modeling is used in this work to indicate a set of equations, derived from the contact principle, that transfer the physical event into the mathematical equations. Continuum rolling contact phenomena is considered to explain how a contact region is formed between rolling die and slab and how the tangential force is distributed over the contact area with coefficient of friction. At the end, elasticity stress behavior of rolling die contact with the slab for a number of cyclic loads is modeled. The model includes new proposed constitutive equations for discontinuity of the velocity–pressure distribution in rolling contact from the entry side to exit side of the neutral point. To verify the model, finite element simulation and experimental data from the literature are considered. The results show good agreement with finite element simulation and experimental data.

**Keywords:** Rolling contact; geometry relationship; pressure distribution; stress behavior

## 1. Introduction

A manufacturing process is the application of engineering industries. It shows how the different problems related to various machine components that may be solved by a study of physical or other governing laws. Continuous casting process is one of the leading process in the metal manufacturing industries. In this process, hot metal stock passes between two rolling dies with plastic deformation when a compressive force is applied from the set of two dies. The two dies have equal size and rotate in opposite directions with the same rotational speed. The space between two rolling gaps is less than the thickness of the entering slab, as shown in the Figure 1. As a consequence, die contact with the slab must be sliding friction in a normal application. Since the friction loads and plastic deformation are the results of interactions between contacting bodies, the nature of contact cannot be understood without explaining the nature of interface information. Modeling of the contact information is one the subjects of Tribology which is a branch of quasi-static phenomena.

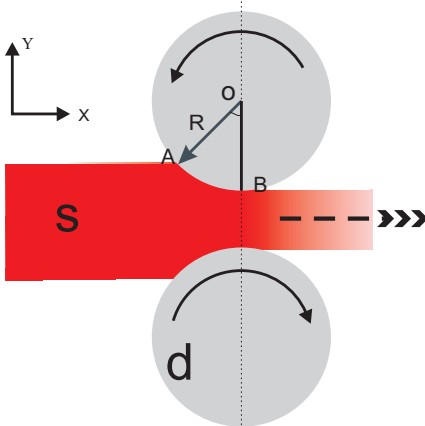

**Figure 1.** Geometrical relation between rolling die and flat rolling.

Metal rolling contact behavior is very complex, and in order to analyse the behavior, investigations have to be made. Rolling process involves contact between the work piece and rolling die. Three main points of views are usually considered in rolling contact, i.e., mechanical, macroscopic and microscopic points of view [1]. To understand rolling contact from the beginning, these points of view are considered mainly from a mechanical point of view by observing the previous works. For instance, wheel railway and automotive tire behavior [2], energy losses in bearings, distortion in a printing press image, profile development [3] and Hertz contact theory. In short, metal forming contact problems will not fall into the preceding categories (Hertzian elastic or frictionless cones/punches) and require their own modeling. The common problems of this sort are complex geometry and nonlinearity of elasticity and plasticity properties of rolling and slab. So, there is no clear situation for explanation. At any time, continuum rolling contact theory must be invoked based on the nature of contact and application.

Extensive research has been conducted regarding process variable relationships during hot rolling and methods to study the physics of slab deformation mechanisms. For instance, work done on modification of roll flattening analytical models based on the plane assumption [4], and a different formulation of the governing equations of flat rolling were developed in Ref. [5,6]. The majority of those studies have been focused on conventional structural physics of slab deformation. In the same way, little research exists on the development of predictive tools that can be applied for rolling die deformation analyses. However, the dominant phenomenon of rolling deformation that occurs in metal forming is basically a complex research topic.

Solid understanding of the behaviour of rolling contact is necessary for successful contact modeling (for further information see [7,8]). Whether the model is predictive or adaptive phenomena in the rolling gap needs clear information. When the two surfaces approach one-another, contact occurs only at discrete points, the total area of which is defined as the true area in the contact. The magnitude and deformation under normal and shear loading are key pieces of information needed to interpret and understand the metal rolling contact mechanisms. One of the first attempts in analyzing the development of the true area of contact is using slip line field analysis of the deforming work piece in [9]. The actions of the asperities in contact and the development of models for friction in metal working are also reported in [10]. Based on experimental results, a numerical analysis to comprehend the stress–strain distribution during the wire drawing process was performed. To achieve this, the same cross sectional area reduction and friction coefficients were used by Sánchez Egea et al. [11]. Varying theories are reviewed in this section from the beginning up to the recent research done in [3,12], in terms of the assumptions and simplifications made either during their development and derivation, or when roll pressures and forces are calculated. It is amazing, no model has been found which is repeatable, and gives better predictions than the others at the level of mathematical approaches. The choice of those models is based on the knowledge of researchers and professionals on the specific domain. Thus, the work proposed in this article is necessary to address such issues. With

the above-mentioned assumptions, as well as those of elastic deformations and other relative issues, the work of this article is proposed for rolling die under a continuous casting process.

## 2. Rolling Contact Problem

The contact problem of rolling dies is the most challenging issue in the metal-forming process due to intrinsically difficulty mathematical formulation problems corresponding to the discontinuity of the velocities in the rolling gap. If the contact information is not taken into account properly, the final results of the computation are greatly affected and can be wrong. When seen from an engineering point of view, this is not straightforward and needs a systematic approach with theoretical background in the field of computational analysis. Most studies on continuum rolling contact are focused on how the physical phenomena proceeds and may neglect the gradual inertia effects. There are certain studies in 2D and 3D contact elastic and plastic dynamics analyzed in the work [13,14].

To start the analysis, let us suppose that the surface of the die is described by function $g(x)$ with coordinate $x$

- $g(x) = 0$ is the surface of the rolling die.
- $g(x) < 0$ is the interior of the die, which is valid for any interior point of the rolling die.

Then the outer normal to the rolling die surface is defined by

$$n = \frac{1}{|\partial g(x)/\partial x|} \cdot \frac{\partial g}{\partial x} \tag{1}$$

To prove the above assumption through mathematical application, let us further consider that two bodies are under motion. The change of the motion with time $t$ is expressed by the velocity $V$. When finite time intervals are considered, approximation of the displacement can be made through velocity formulation. In the same way, it is possible to assume that the die is in the contact with slab for which the domain $Q$ can be represented by a function $g(x,t)$, i.e., as a function of the coordinate $x$ and $t$ can be explained

$$g(x,t) = 0 \tag{2}$$

At time $t$, the slab point coordinate $x'(t)$ is on the rolling die surface if

$$g[x'(t),t] = 0 \tag{3}$$

And at the time $t + dt$ the same slab point on the die surface has moved such that the new coordinate vector $x'(t + dt)$ will satisfy

$$g[x'(t+dt), t+dt] = 0 \tag{4}$$

Which is transformed with a first-order Taylor expansion and gives

$$g\left[x'(t+dt), t+dt\right] \cong g\left[x'(t),t\right] + \frac{\partial g}{\partial x}\left[x'(t),t\right]\frac{dx'}{dt} + \frac{\partial g}{\partial t}\left[x'(t),t\right]dt = 0 \tag{5}$$

In short, Equation (5) can be replaced by

$$x'(t+dt) = x'(t) + dt \cdot V(t+dt) \tag{6}$$

This allows us to update the domain $Q$ into $x'(t + dt)$ and to continue the calculation. Due to its simplicity, this scheme obviously helps in better understanding as shown in Figure 2. They have the same location at time $t$, that is $x(t) = x'(t)$, and the velocity change $(\Delta V)$ is given by

$$\Delta V = V_s - V_r \tag{7}$$

where $V_s$ is velocity of the slab and $V_r$ is velocity of the rolling die.

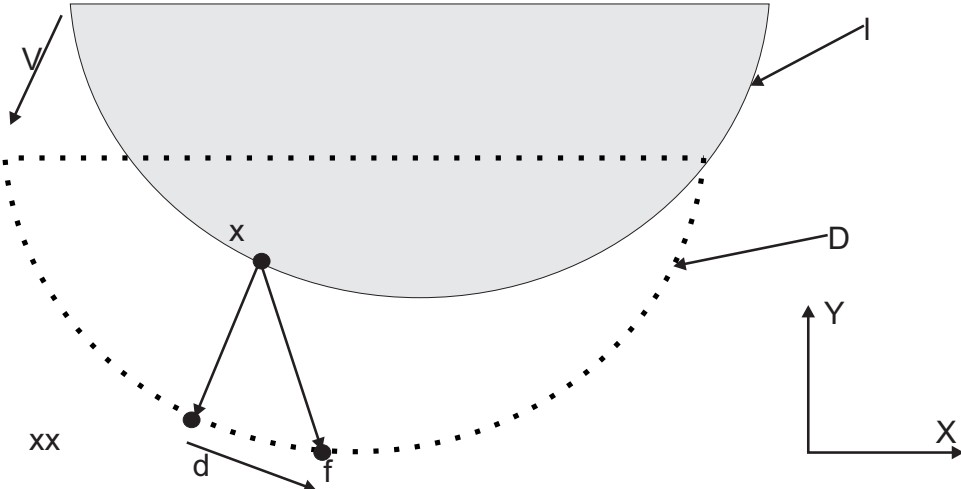

**Figure 2.** The slab point and rolling die surface have same position at the time *t* but at the increment of time $t + dt$ they have different position due to the velocity difference.

For general curved surfaces, approximated finite incremental times are considered, but Equation (7) is not enough to explain all possibilities of physical situations that can occur in reality. This is because it does not allow the slab points to penetrate into the die or loss contact. Figure 3 shows slab gliding on the rolling die and this is mathematically expressed as

$$\text{if} \quad \sigma_n = (\sigma \cdot n) \cdot n < 0 \qquad \text{then,} \quad \Delta V \cdot n = 0 \tag{8}$$

where $\sigma$ is compressive stress. This means that the slab point is already on the contact surface at the beginning of the increment. Then the slab point must remain in contact with the die, and due to the curvature of the die at the end of increment, the point exhibits a small departure from the die surface but it must be reprojected

$$\text{if} \quad \sigma_n = (\sigma \cdot n) \cdot n = 0 \qquad \text{then,} \quad \Delta V \cdot n \geq 0 \tag{9}$$

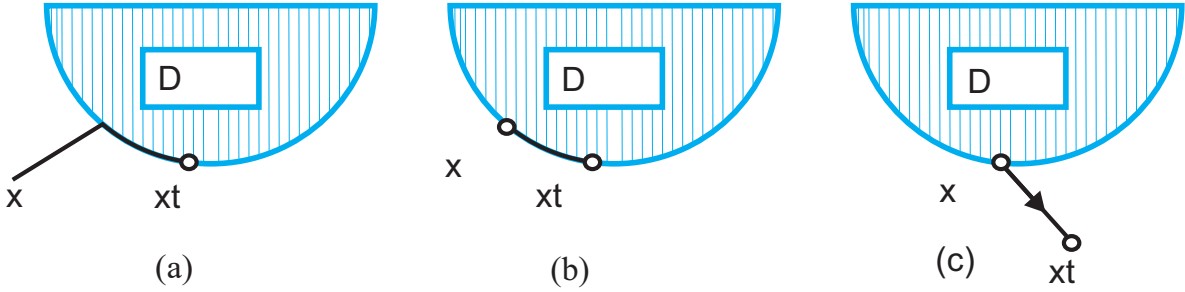

**Figure 3.** Incremental contact the rolling die is assumed that fixed. (**a**) Contact establishment, (**b**) Gliding contact, and (**c**) Loss of contact.

The slab point is on the free surface at the beginning of the increment, and penetrates inside the die at the end of the increment. It is possible to orthogonally reproject the point with coordinates $x(t + t)$ on the surface of the die by reducing the time increment such that it is possible to choose the minimum time $t^*$ for all the points coming into contact. There is an accurate technique for further analysis, which may increase the number of time increments significantly. This technique is mainly used in three-dimensional analysis. Using the three main incremental point contacts with the level of implicitly schemes, the above-discussed issues can be summarize and presented as follows:

1.　The slab point and die are free surfaces at the beginning of the increment and come into contact during the increment.
2.　Slab point and die bodies are already in contact at the beginning of the increment and the normal compressive stresses are induced at the beginning and at end of the increment, where the two bodies are remain in the contact.
3.　The slab point is in contact at the start of the increment, but the normal stress becomes zero, then the slab is allowed to leave the die surface.

## 3. Tribology of Metal Rolling and Geometry Relationship

The phenomena of contact between rolling dies and slab can be explained in terms of the friction hypothesis which examines the origin of the resistance to relative motion in terms of slid or stick between the two contacting surfaces. The hypothesis, presented in [15], which is credited to the French scientist Desaguliers, explains the origin of resistance to motion in terms of adhesive bonds. The hypothesis states that the surface of engineering materials are generally never completely smooth, and they contain asperities and valleys when viewed under suitable magnification. In metal rolling, friction is responsible for drawing the slab into the rolling gap.

Friction is a phenomenon of shearing stress when it acts in a tangential way to the rolling dies at any section along the arc of contact with the slab. However, the direction of shearing stress reverses at the neutral point [16,17]. Between the entry section of the rolling gap and the neutral section, the direction of friction is the same as the direction of motion of the slab. Therefore, the friction aids in pulling the slab into the rolling gap as part of the travel. The direction of friction reverses after the neutral point when the velocity of the slab is higher than the velocity of the rolling die. Friction force opposes the forward motion of the slab in beyond the neutral point. In the same way, rolling die pull forward (see Figure 4). This interpretation, magnitude of the friction acting ahead of neutral point is greater than that beyond the neutral point. Therefore, the net friction is acting along the direction of the slab movement, thereby aiding the pulling of the slab forward.

Rolling die exerts a normal pressure on the slab which is imagined to be the pressure exerted by the slab on the dies to separate them. Frictional shear stress is induced due to friction force in the tangential direction *F* in the rolling gap. This relationship is shown in Figure 5 with rolling die pressure distribution $P(r, \theta)$ and is mathematically expressed as follows

$$F = \mu P(r, \theta) \tag{10}$$

where $\mu$ is coefficient of friction, *P* is a function of variables *r* and $\theta$. In this case, *r* and $\theta$ are the coordinates in a die (cylindrical) system. In the other word, *r* and $\theta$ are indicates the direction of the loads in the radial and/or circumferential direction, respectively. Based on the variations in application conditions, the slip coefficient friction is considered between the rolling die and slab. At the entry section, i.e., if the forces acting on the slab are balanced, Equation (10) can be formulated

$$P(r, \theta)sin\alpha = \mu P(r, \theta) \tag{11}$$

The slab is pulled into rolling at entry section without considering that the slab bulges out in the transversal direction. Such a problem may even happen in continuous casting at an early strand of deformation. For the slab to be drawn into the rolling gap, the following condition should be satisfied

$$\mu P(r, \theta)cos\alpha \geq P(r, \theta) \sin \alpha \tag{12}$$

$$\mu \geq \tan \alpha \tag{13}$$

In other words, if the tangent of the angle of bite exceeds the coefficient of friction, the slab will not be drawn into the rolling gap. The detailed mathematical expression can be derived by considering

the necessary conditions. i.e., angle of bite or angle of contact, slip direction in arc contact and neutral point, as shown in Figure 4.

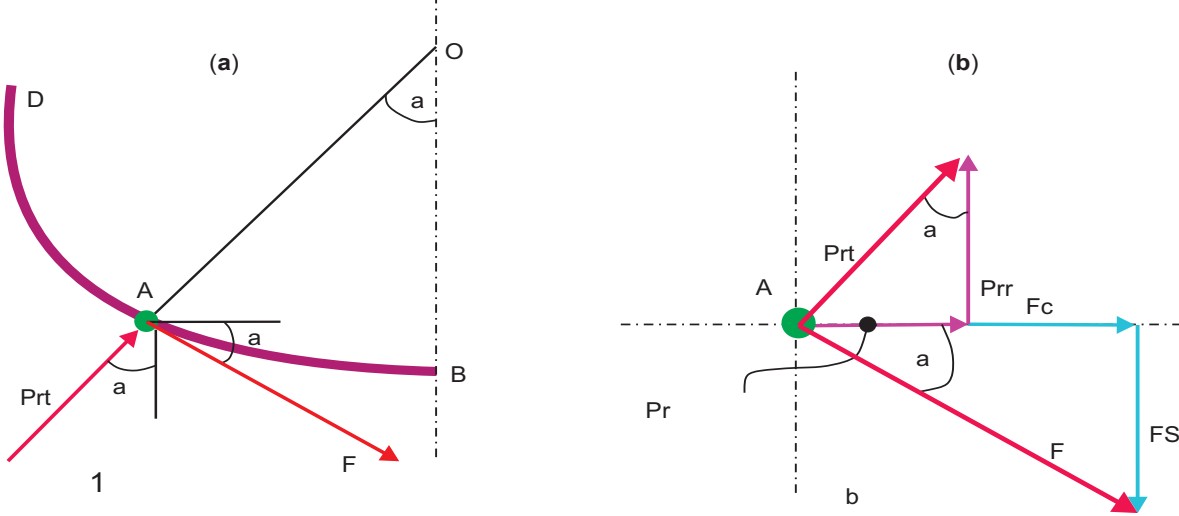

**Figure 4.** Action of friction and normal force in the contact area.

**Figure 5.** Component force. (**a**) Direction of the two forces, (**b**) Resolved force that act on rolling die.

Before proceeding to determine the relationship between force and geometry in rolling contact, first let us consider the plastic deformation of the slab direction. According the condition given in Equations (12) and (13) the result of incremental width is zero. Thus, the vertical compression of the metal is translated into an elongation in the rolling direction. As the slab is dragged by the dies into the rolling gap, it decreases in thickness while passing from the entrance to the exit. From the volume metal conservation law, one can write the following equation

$$wh_0V_0 = whV = wh_fV_f \tag{14}$$

where $w$ is the width of deformed slab, $V$ is velocity at any thickness and $h$ is intermediate between initial thickness $h_o$ and final thickness $h_f$. At the mean time slab, velocity gradually increases from $V_0$ at the entrance to $V_f$ at the exit. In order that vertical elements in the metal remain undistorted, Equation (14) requires that the exit velocity $V_f$ must be greater than the entrance velocity $V_0$. Due to the variation in the velocities of die surface and slab surface, there will be relative motion between them. The relative motion between the die and the slab is referred to as a slip. Slip takes place at the front and the back of the rolling dies in the rolling gap. There is a point where the velocity of $V_r$ equal to $V_s$, which is called neutral point. It is indicated in Figure 4 by angle $\alpha_n$. Before determining the neutral point, let us find angle of contact $\alpha$. The angle $\alpha$ between the entrance plane and the centerline of the rolling die as shown in Figure 6. From the definition of circular segment and using Taylor's series expansion $\alpha$ can be derived

$$h = h_f + 2R\left(1 - cos\alpha\right)$$

$$cos\alpha = 1 - \frac{\alpha^2}{2!} + \frac{\alpha^4}{4!}...$$

$$h = h_f + R\left(\alpha^2\right)$$

$$cos\alpha = 1 - \frac{\Delta h}{2R} \tag{15}$$

where, $\Delta h = h_0 - h_f$.

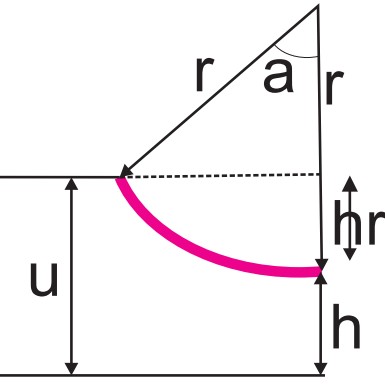

**Figure 6.** Analysis of angle contact bite with a given radius of rolling die.

The position of the neutral angle (the pseudo independent parameter) is also determined as one of independent function parameters from Figure 4 and rearranging the equation that was developed in [18] and considering hot rolling conditions as well as other analyses based in [19]. The rest of the analytical treatment is presented in this section with symmetry assumption. Top and bottom are mirrored around the plane of symmetry. So, the pressure distribution and the position of the neutral

point are identical for the top and bottom or left and right side. Taking the above justification into account, the location of the neutral point is calculated by

$$\alpha_n = \sqrt{\frac{h_f}{R}} \tan\left(\frac{H_n}{2}\sqrt{\frac{h_f}{R}}\right) \tag{16}$$

where, $H_n = \sqrt{\frac{h_f}{R}} - \frac{1}{2\mu} ln \sqrt{\frac{h_f}{R}}$

In the same way, flattening neutral point due to rolling die flattening radius $R'$ can be given

$$\alpha'_n = \sqrt{\frac{h_f}{R'}} \tan\left(\frac{H_n}{2}\sqrt{\frac{h_f}{R'}}\right) \tag{17}$$

where R' is determined from Hitchcock's equation [20]

$$R' = R\left[1 + \frac{cP'(r,\theta)}{w(h_0 - h_f)}\right] \tag{18}$$

where, $c = 16\left(1 - v^2\right)/\pi E_r$, $P'(r,\theta)$ is pressure distribution with flattened die and $E_r$ is young's modulus of the die material.

## 4. Pressure Distribution in Contact Area

The shape of the die–slab contact surface is not known at the exact point. Computation and analysis of pressure distribution must be combined with the deformed slab and rolling die by suitable equation. A number of theories and equations were attempted in earlier studies, but there are no rationalized differences between them or no model has been found which is repeatable, and gives better predictions than the others. For instance, Bland and Ford's model [21], Sims' model [22], Alexander's model [23], refinement of the Orowan models [24] and new cold-rolling theory [25]. However, most of the approaches offered some unique advantage and also had deficiencies of one sort or another. On the other hand, the choice of those models is based on the knowledge of researchers and professionals on the specific domain.

Hot-rolling has no advance to the state of knowledge that exists for cold-rolling due to the problem of inhomogeneous deformation and less well defined friction conditions. In hot-working conditions, the flow stress is also a function of temperature and speed of rolling die. With the above-mentioned plasticity/elastic deformations and other relative issues after considered, pressure distribution equation between rolling die and slab is proposed. For this approach the following assumptions are considered, that is the same state of stress exists at all points of the plane $x$, $y$ and $z$ directions. In the same way, considering a volume element of unit width having the instantaneous thickness change $dh$ and length $dx$ (see Figure 7). Furthermore, assume that the longitudinal stress $\sigma_x$, which is constant over a plane normal to the $x$ direction, and the pressure $\sigma_{yy} = -P(r)$ acting on rolling die interface, are the two principal stresses which are acting. In this case, $P(r)$ is a pressure distribution in radial direction (in $Y$-axis direction) as shown in Figure 7. As a result of these assumptions, a slab is deformed in the plane.

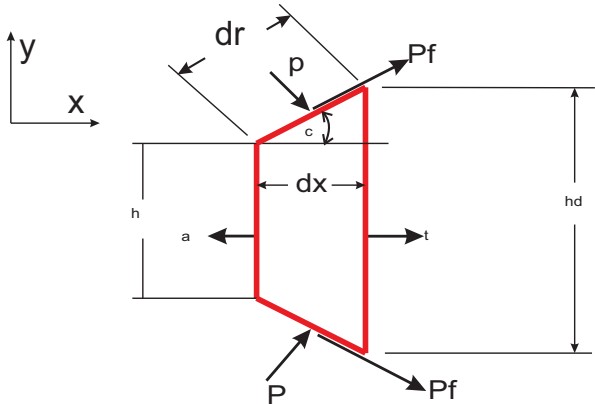

**Figure 7.** Loads on slab element.

Taking the equilibrium of forces in the vertical direction results in a relationship between the normal pressure and the radial pressure. The relationship between the normal pressure and the horizontal compressive stress $\sigma_x$ is given by the distortion energy criterion of yielding in plane strain. Because of symmetry, the metal flows away from the center plane in both directions, and it will be sufficient to analyze the conditions for half side. By introducing the shearing stresses, $F = \tau_{xy}$ acting on the surface, the equilibrium of the $x$ component of the forces acting on a volume element is expressed by

$$hd\sigma_x + \sigma_x dh + 2P(r)dx(\tan \alpha + \mu) = 0 \tag{19}$$

The distortion-energy condition of plasticity is expressed, in the case of plane strain, by the relationship

$$\sigma_x - \sigma_{yy} = \frac{2}{\sqrt{3}}\sigma_0 = \sigma_0', \qquad \sigma_x = \sigma_0' - P(r), \quad d\sigma_x = -dP(r) \tag{20}$$

where $\sigma_0$ is yield strength and $\sigma_0'$ is deviatory stress. Substitute Equation (20) into the Equation (19)

$$-hdP(r) + \left(\sigma_0' - P(r)\right)dh + 2P(r)dx(\tan \alpha + \mu) = 0 \tag{21}$$

By considering that

$$dx = \frac{dh}{2\tan \alpha} \tag{22}$$

Then, one can get

$$hdP(r) - \left(\frac{\mu P(r)}{\tan \alpha} + \sigma_0'\right)dh = 0 \tag{23}$$

Introducing the parameter $w'$, which is defined by the equation

$$\tan w' = \sqrt{\frac{R}{h_f}}\tan \alpha \tag{24}$$

Following the procedure of Equation (15)

$$h = h_f(1 + \tan w') \tag{25}$$

Its derivative with respect to the parameter $w'$ is

$$\frac{dh}{dw'} = 2h_f \tan w'(1 + \tan w') \tag{26}$$

Substituting the expressions from Equations (25) and (26) into Equation (23) and introducing the the constant

$$\phi = 2\mu \sqrt{\frac{R}{h_f}} \tag{27}$$

One has differential equation and multiplying both sides by $e^{\phi w'}$

$$e^{\phi w'} \left( dP(r) - \left( P(r)\phi + 2\sigma'_0 \tan w' \right) dw' \right) = 0 \tag{28}$$

At the end, rearranging and then integrating:

$$P(r) = e^{\phi w'} \left( 2\sigma'_0 \int e^{-\phi w'} \tan w' dw' + C_1 \right) \tag{29}$$

where $C_1$ is constant of integration. According to relation given in Ref. [26], the integral appearing in this equation can be calculated with sufficient approximation by putting $\tan w' \approx w'$, then

$$\int e^{-\phi w'} \tan w' dw' = -\frac{e^{-\phi w'} \left( 1 + \phi w' \right)}{\phi^2} \tag{30}$$

The general solution of this equation

$$\frac{P(r)}{\sigma'_0} = Ce^{-\phi w'} - \frac{2}{\phi^2} \left( 1 - \phi w' \right) \tag{31}$$

From above expression

$$\frac{\sigma_x}{\sigma'_0} = 1 - \frac{P(r)}{\sigma'_0} = 1 - Ce^{-\phi w'} - \frac{2}{\phi^2} \left( 1 - \phi w' \right) \tag{32}$$

At the boundary condition of entry side (just at outside of entry point), $(\sigma_x)_{x=0} = 0$, and $C$ can be obtained from Equation (32) with this condition. And then, furnish the above expression with all situations, the pressure distribution on the entrance side of the neutral point can be given as

$$\frac{P(r,\theta)}{\left( 2/\sqrt{3} \right) \sigma_0} = e^{\phi(w'_0 - w')} + \frac{2}{\phi^2} \left( e^{\phi(w'_0 - w')} (1 - \phi \cdot w'_0) - (1 - \phi \cdot w') \right) \tag{33}$$

The pressure distribution on the exit side of the neutral point is expressed as follows

$$\frac{P(r,\theta)}{\left( 2/\sqrt{3} \right) \sigma_0} = e^{\phi \cdot w'} + \frac{2}{\phi^2} \left( e^{\phi \cdot w'} - (1 + \phi \cdot w') \right) \tag{34}$$

where

$$w' = \tan^{-1} \left[ \sqrt{\frac{R}{h_f}} \tan \alpha_n \right], \quad w'_0 = \tan^{-1} \left[ \sqrt{\frac{R}{h_f}} \tan \alpha \right] \tag{35}$$

The magnitude of the coefficient of friction at the die–slab interface has been commonly characterized in two ways. The first method of characterizing interface friction by the use of a constant coefficient of friction (Coulomb friction). The second method of calculating interface coefficient of friction assumes that the interface zone may be represented by material constant, in which friction factor is constant in the rolling gap. However, the phenomena of hot metal-rolling includes the frictional events like heat transfer and other related parameters. In other words, the frictional stress

instantly changes direction at the neutral point, which is identified based on an initial guess for the velocity field. From this justification, the following equations are considered from the literature

$$\mu = 1.05 - 0.0005T - 0.056V_r \tag{36}$$

$$\mu = -1.607 - 0.13V_r + 1.256 \tag{37}$$

$$\mu = \frac{\Delta h / R'}{2\sqrt{\frac{\Delta h}{R'}} - 4\sqrt{\frac{2S_f h_f}{2R' - h_f}}} \tag{38}$$

where $T$ is the temperature, given here in $^\circ C$ and $V_r$ is given in $m/s$. For steel rolling, the relevant formula for $\mu$ in Equation (36) is developed by Geleji in [27] and even cited by different researchers. In the same way, Equations (38) and (37) are given in [28] and [29], respectively. Where the $\mu$ depends on the surface velocity of the rolling die, reduction and temperature is shown in Figure 8. The result shows that $\mu$ decreases with increasing rolling die speed and temperature, at the same time, it is affected by reduction ratio.

In Figure 9, two lines are intersected at the neutral point due to the action of compression and tension force where the model is derived from Equations (33) and (34). For this method, $R = 100$ mm, $h_f = 40$ mm, $T = 500\,^\circ C$ and $V_r = 5$ m/s are considered, the result indicates that tension force is greater than the compressive load, which means maximum stress is exerted on the rolling die from exited side rather than the entry side of neutral point. In the same way, FORTRAN program was developed with PYTHON script to simulate this phenomena in ABAQUS 6.14 (ABAQUS Inc., Palo Alto, CA, USA), as depicted in Figure 10, where the following steps are considered:

1. H13 tool steel material properties are considered for the rolling die where the data can be found in [30]
2. A36 mild steel material properties are considered for the slab where data are available in [31]
3. After material properties are collected, constitutive equations are implemented in ABAQUS via UMAT with following steps:

   - Proper definition of constitutive model,
   - Definition of state variables,
   - Transforming the constitutive equation that inform of incremental through proper integration scheme,
   - Calculation of the Jacobian,
   - At the end, connect the written code to the ABAQUS subroutine,
   - In addition, to exclude the slab deformation result from the general solution, Bens.exe and Zedgrapp.dll should be executable with ABAQUS.

More details of these steps along with illustrative examples can be found in [32]. However, in ABAQUS/Explicit the user-defined material model is implemented in user subroutine VUMAT. The procedure is the same as UMAT where the result shown in Figure 10 for which Dynamic/Explicitly steps are considered. UMAT and VUMAT are used when none of the existing material models included in the ABAQUS material library accurately represent the behavior of the material to be modeled. Multiple user materials can be implemented in a single UMAT or VUMAT routine and can be used together.

Figure 9 shows analytical results of maximum stress distribution, which is induced in rolling die at the entry and exit sides. On the other hand, finite element simulation agree with the numerical computation that are implemented in python along the arc of contact when plasticity deformation is performed. However, the numerical model shows exaggerated stress at the specific point which means maximum tension stress developed due to the maximum velocity of slab from exit side being high.

Finally, the result suggests that tension load exerted on rolling die is different from the entry side to exit side in magnitude and direction.

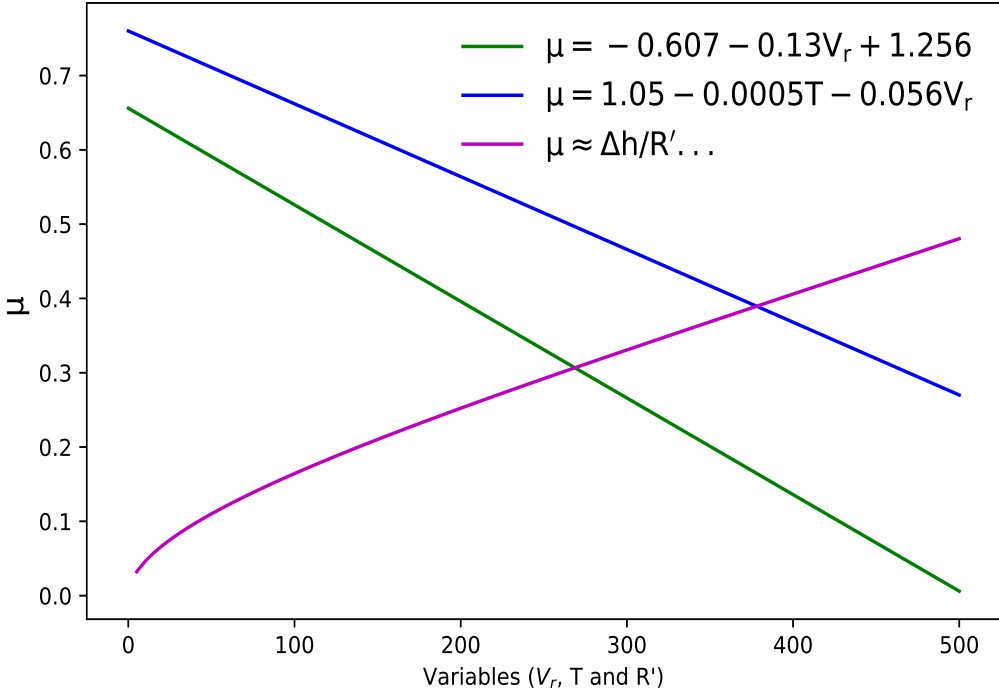

**Figure 8.** The coefficient of friction as a function of the temperature, reduction and the rolling die surface velocity.

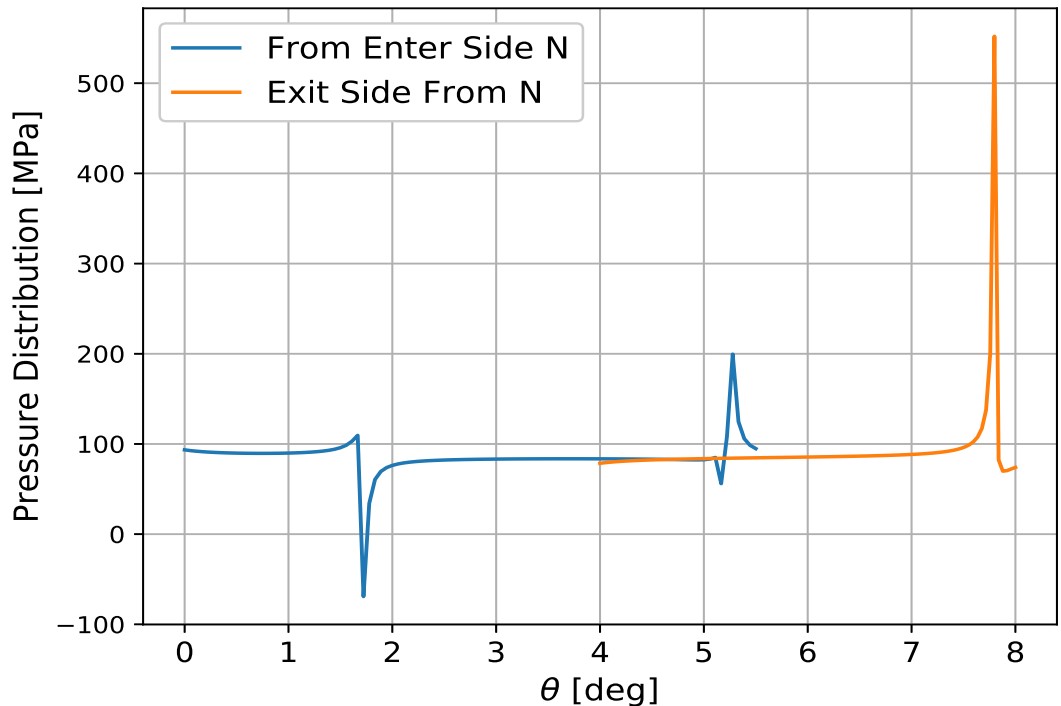

**Figure 9.** Pressure distribution phenomena under rolling contact from entry side and the exit side of the neutral point versus the bite angle.

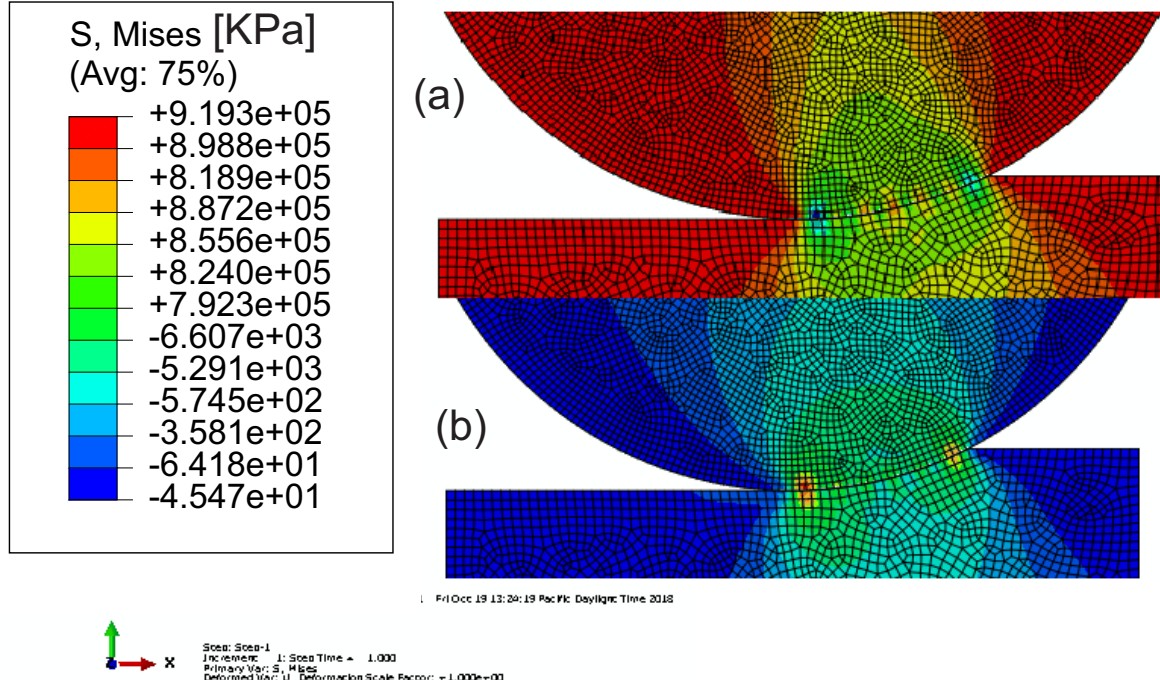

**Figure 10.** Stress distribution along the arc of contact during plasticity deformation in the global section view where the results from ABAQUS. (**a**) Radial stress distribution, (**b**) Tangential stress distribution.

To get a unified equation, let us find the mean rolling die pressure at interface through integration. Considering a volume element of unit width and having a thickness $\tilde{h}$ with length $dx$ (see Figure 11).

$$\tilde{h}d\sigma_x - 2\tau_{xy}dx = 0 \tag{39}$$

Which is obtained by substituting Equation (20) in the Equation (39), and then separating the variables. In addition, it is known that the relationship of Coulomb's law in slide friction is given by, $\tau_{xy} = \mu P(r)$

$$dP(r) + 2\frac{\mu P(r)}{\tilde{h}} = 0, \qquad \frac{dp(r)}{P(r)} = -2\frac{\mu dx}{\tilde{h}} \tag{40}$$

Integrating both sides

$$\ln P(r) = -\frac{2\mu x}{\tilde{h}} + C \qquad \text{or} \qquad P(r) = C_1 e^{-2\mu x/\tilde{h}} \tag{41}$$

where $C_1 = e^c$ is a new integration constant to be determined from the boundary condition at $x = L_c/2$, $\sigma_x = 0$ and consequently $P(r) = \sigma_0'$, then obtain

$$C_1 = \sigma_0'e^{\left(\mu L_c/\tilde{h}\right)} \tag{42}$$

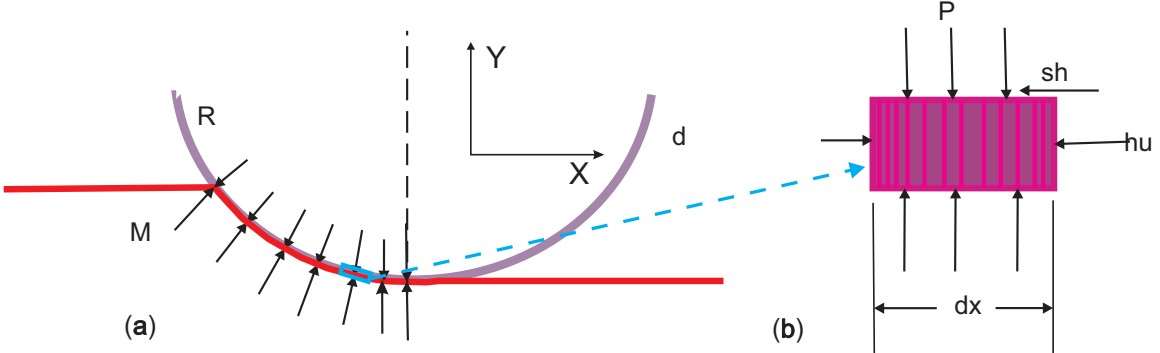

**Figure 11.** Load relationship under rolling contact. (**a**) Load acting on slab/die, (**b**) Dimension of a volume element of plate in rolling contact and stresses acting on volume element.

After substitution, the mean pressure is

$$\tilde{P} = \int_0^{L_c} \frac{P(r)dx}{L_c} = \sigma_0'\frac{e^{\mu L_c/\tilde{h}} - 1}{\mu L_c/\tilde{h}} \tag{43}$$

where, $\tilde{h} = (h_0 + h_f)/2$ and $L_c$ is the length of contact between the slab and rolling die as shown in the Figure 4. In Figure 12 computational result of mean pressure is compared with the Equations (44) and (45), which is purposely developed for hot working condition in [33] and [34], respectively

$$\tilde{P} = \sigma_0'\left(1.31 + 0.53\frac{L_c}{\sqrt{h_0 h_f}}\right) \tag{44}$$

$$\tilde{P} = \sigma_0'\left(\frac{\pi}{2} + \frac{L_c}{h_o + h_f}\right) \tag{45}$$

In Figure 12, where reduction ratio and roll diameter are constant, the angle contact length is the relative distance along $x$. The same basic equation for hot-rolling with important condition which were

given for determining the effective pressure in rolling gap. Upon comparison, the difference between the models appeared as per the authors' concern. Accordingly, the Ford model shows maximum mean pressure in rolling gap, whereas the Deston model shows minimum pressure load. In the same way, the model proposed in this work is relatively the average of the two models. In the continuous casting process, the proposed model is fair enough because its main objective is supporting highly heated metal for further subsequent strands rather than breaking down the casting ingot into slab/bloom under single strand. However, for given $L_c/h$, an increasing coefficient of friction easily leads to more deformation pressure. The role of $\mu$ in raising pressure distribution should be clear from the above calculation. Note also that the role of $\mu$ becomes particularly important at large values of $L_c/h$.

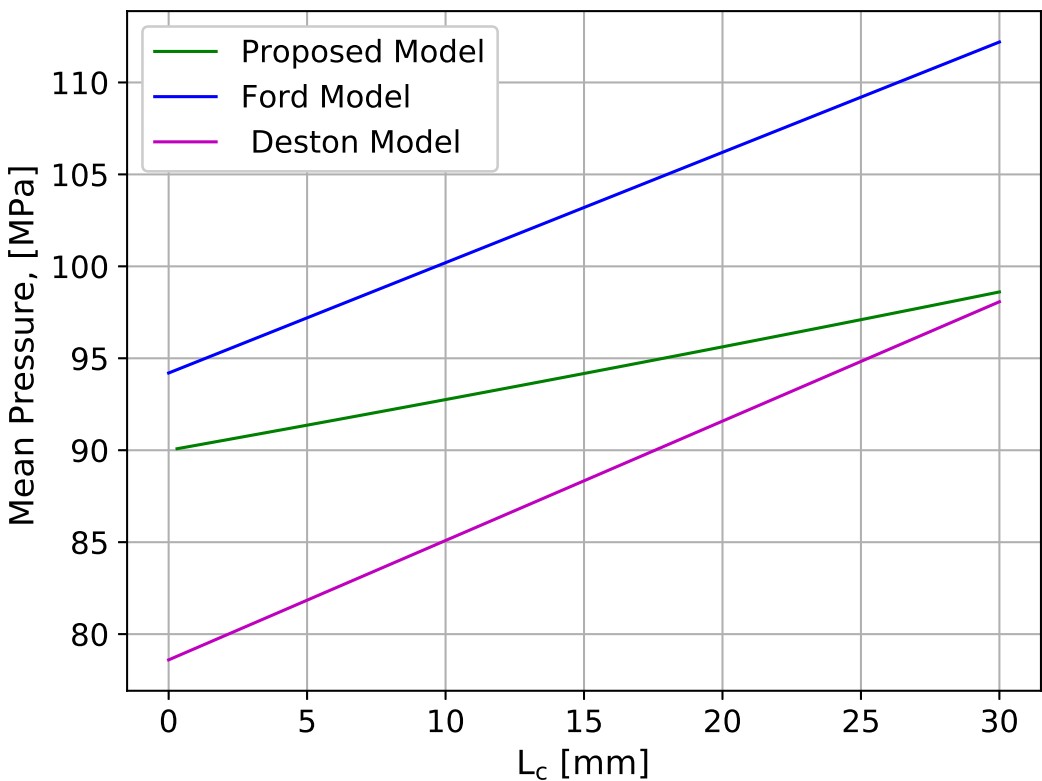

**Figure 12.** Mean pressure versus contact length.

## 5. The Deformation of the Rolling Die and Constitutive Equation

The review of Robert in [16] concerning the Hitchcock's formula that is given in [20] to calculate the radius of the flattened is not accepted, as mentioned in Section 3. Rolling die deformation modeling is the core model of this section which can determine stress behavior under rolling contact for a number of cyclic loads $n$. Numerical modeling is a hotspot of research aiming to tackle experimentally tedious tasks and expenses. Thus, analysis and modeling play a vital role in metal-forming and tool design industries.

The problem of the deformation of the rolling die is treated here by assuming that a solid cylinder is subjected to non-symmetrical loads. The schematic loading diagram of a rolling die is shown in Figure 13. The loading consists of the radial stress and the interfacial shear stress, where the radial stress is designated by $\sigma_{rr}$ and the interface shear stress by $\tau_{r\theta}$. These loads must be balanced by a statically equivalent force system, the location of which would depend on the type of mill considered. For four-high mills or six-high mills, the backup rolls would supply the necessary balance, while for two-high mills the reaction forces would be applied through the bearings of the rolling die journals.

The present analysis is concerned with a two-dimensional treatment of the rolling die of a two-high mill (usually used in continuous casting) and die bending being neglected. The balancing loads are taken to be distributed over the dies with $2\xi = \pi$, where $2\xi$ is the extent of the pressure distribution of the now imaginary back-up roll instead of journal bearings. This condition ensures that the mid planes of the rolling die remain undeformed and stationary.

The stress distribution in any problem of linear elasticity should satisfy the biharmonic equation, which in $2D$ cylindrical coordinates is

$$\left( \frac{\partial}{\partial r^2} + \frac{1}{r}\frac{\partial}{\partial r} + \frac{1}{r^2}\frac{\partial^2}{\partial \theta^2} \right) \left( \frac{\partial^2 \varphi}{\partial r^2} + \frac{1}{r}\frac{\partial \varphi}{\partial r} + \frac{1}{r^2}\frac{\partial^2 \varphi}{\partial \theta^2} \right) = 0 \tag{46}$$

where the coefficients of terms singular at the origin were taken to equal zero. The radial and shear stresses are then obtained terms of the Airy stress function

$$\sigma_{rr} = \frac{1}{r}\frac{\partial \varphi}{\partial r} + \frac{1}{r^2}\frac{\partial^2 \varphi}{\partial \theta^2} \tag{47}$$

and

$$\tau_{r\theta} = \tau_\theta = -\frac{\partial}{\partial r}\left[ \frac{1}{r}\left( \frac{\partial \varphi}{\partial \theta} \right) \right] \tag{48}$$

Following the procedure of [35], the stress and strain distributions in the rolling die in a state of plane strain may be calculated from a stress function by using biharmonic functions

$$\varphi = c_0 r^2 + d_1 r^3 \sin\theta + d_2 r^3 \cos\theta + \sum_{n=2}^{\infty} \left( a_{1n} r^n + b_{1n} r^{n+2} \right) \sin(n\theta) \; + $$
$$\left( a_{2n} r^n + b_{2n} r^{n+2} \right) \cos(n\theta) \tag{49}$$

where the constants $a_{1n}$, $a_{2n}$, $b_{1n}$, $b_{2n}$, $c_0$, $d_1$ and $d_2$ need to be determined such that the stress boundary conditions at $r = R$ are satisfied. They are determined next by representing the normal and shear loading on the rolling die's surface

$$\sigma_{rr} = P(r) \qquad \text{and} \qquad \tau_{r\theta} = \tau(\theta) \tag{50}$$

Normal and shear loading on the rolling die's surface in terms of fourier series

$$P(r) = P_{a0} + \sum_{n=1}^{\infty} \left[ P_{an} \cos(n\theta) + P_{bn} \sin(n\theta) \right] \tag{51}$$

and

$$\tau(\theta) = \tau_{a0} + \sum_{n=1}^{\infty} \left[ \tau_{an} \cos(n\theta) + \tau_{bn} \sin(n\theta) \right] \tag{52}$$

where the coefficients are obtained from the Euler formula. For the normal pressure distribution, then

$$P_{a0} = \frac{1}{2\pi} \int_{a_1}^{a_2} P(\theta)\, d\theta + \int_{-\pi/2}^{\pi/2} R_b(\theta)\, d\theta \tag{53}$$

$$P_{an} = \frac{1}{\pi} \int_{a_1}^{a_2} P(\theta)\cos(n\theta)\, d\theta + \int_{-\pi/2}^{\pi/2} R_b(\theta)\cos(n\theta)\, d\theta \tag{54}$$

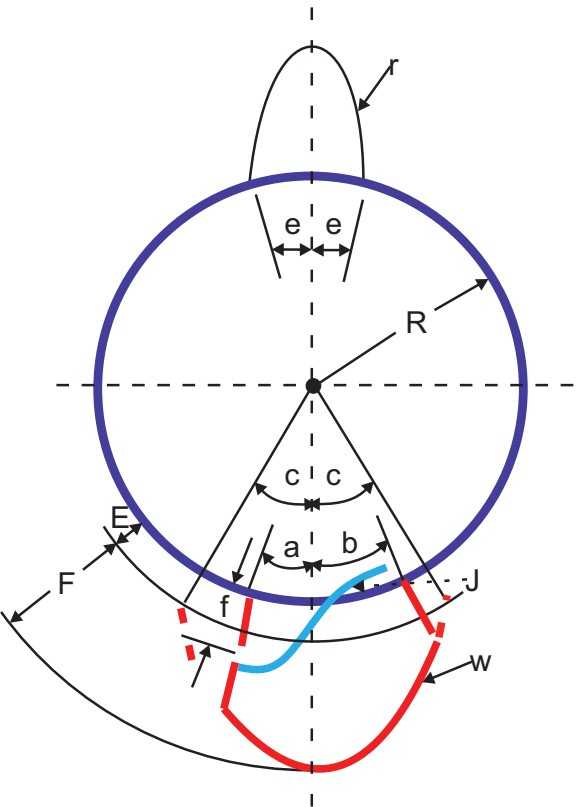

**Figure 13.** The loading diagram of the rolling die showing the rolling pressure and the interfacial shear stress distributions in addition to the forces that keep the die in equilibrium, for further information see [36].

and

$$P_{bn} = \frac{1}{\pi} \int_{a_1}^{a_2} P(\theta) sin(n\theta) d\theta + \int_{-\pi/2}^{\pi/2} R_b(\theta) sin(n\theta) d\theta \tag{55}$$

And for the shear stress distribution

$$\tau_{ao} = \frac{1}{2\pi} \int_{a_1}^{a_2} \tau(\theta) d\theta \tag{56}$$

$$\tau_{an} = \frac{1}{2\pi} \int_{a_1}^{a_2} \tau(\theta) cos(n\theta) d\theta \tag{57}$$

and

$$\tau_{an} = \frac{1}{2\pi} \int_{a_1}^{a_2} \tau(\theta) \sin(n\theta) d\theta \tag{58}$$

where the values of $a_1$ and $a_2$ depends on the neutral point and amplitude of the reaction force, as shown in Figure 13. The rolling die pressure distribution in the roll gap can also expressed as a cosine function using optimization methods

$$P(r) = -E - F cos \frac{\pi\theta}{2\beta} \tag{59}$$

where $E$ and $F$ are constants. The function $R_b$ in Equations (53)–(55) represents the reactions required to keep the rolling die in equilibrium. This can be calculated by expressing the reactions using Fourier series.

$$R_b(\theta) = R_0 + \sum_{n=1}^{\infty} R_{an} \cos(n\theta) + R_{bn} \sin(n\theta) \tag{60}$$

where the coefficients are obtained by the Euler formula

$$R_0 = -\frac{1}{\pi^2} R_m \xi \tag{61}$$

$$R_{an} = -2\xi R_m \frac{\cos[(-\pi + \psi + \xi)n] + \cos[(-\pi + \psi - \xi)n]}{\pi^2 - 4\xi^2 n^2} \tag{62}$$

$$R_{abn} = -2\xi R_m \frac{\sin[(-\pi + \psi + \xi)n] + \sin[(-\pi + \psi - \xi)n]}{\pi^2 - 4\xi^2 n^2} \tag{63}$$

In the above equations, $\psi$ is the angle between the resultant reaction force and the vertical axis, $R_m$ is the amplitude of the reaction force and $\xi$ represents half of the angle over which reaction $R_b$ is distributed. It is noted that, beyond two mills, the value of $\xi$ may be determined from the Hertz contact stresses. However, as was mentioned above, for a two-high mill, the reactions are represented by letting $\xi = \pi/2$ which in fact indicates the distribution of those forces across a diametral plane of the rolling die. In Equation (52) the shear stress distribution can be is represented by

$$\tau(\theta) = F' \sin\frac{\pi\theta}{2\beta} \tag{64}$$

$$a_{1n} = \frac{nP_{bn} - (n-2)\tau_{an}}{2n(n-1)r_b^{n-2}}, \qquad a_{2n} = \frac{nP_{an} - (n-2)\tau_{bn}}{2n(n-1)r_b^{n-2}} \tag{65}$$

$$b_{1n} = \frac{P_{bn} - \tau_{an}}{2n(n-1)r_b^{n-2}}, \qquad b_{2n} = \frac{P_{bn} - \tau_{an}}{2n(n-1)r_b^{n}} \tag{66}$$

$$d_1 = \frac{P_{b1}}{2r_b} = -\frac{\tau_{a1}}{2r_b}, \qquad d_2 = \frac{Pa_1}{2r_b} = \frac{\tau_{b1}}{2r_b} \tag{67}$$

$$c_0 = \frac{P_{a0}}{2} \tag{68}$$

To determine the strain components one can use the plane strain form of hook's law

$$\varepsilon_{rr} = \frac{1+v}{E_r}\left[(1-v)\,\sigma_{rr} - v\tau_\theta\right], \qquad \varepsilon_{\theta\theta} = \frac{1+v}{E_r}\left[-v\sigma_{rr} + (1-v)\,\tau_\theta\right] \tag{69}$$

The comparison between the numeric and finite element simulation result with experimental data is presented in Figure 14. In the same way, finite element computation is modeled as shown in Figures 15 with following the steps that explained in the Section 4. Where the initial load 90 MPa is considered for numeric and FEM computation, the result shows good overlapping. At the same time, both computational results are compared with experimental data, where the result shows good agreements as depicted in Figure 14. Meanwhile, a computer program is written to compute the system of Equation (52) for various values of the input parameters, including 3D modeling for surface folding due to tension and compressive loads on the surface of the rolling die as shown in Figure 16 and 17, respectively. At the end, overall radial and tangential stress for a number cyclic loads in rolling gap is modeled and the result is shown in Figure 18.

A critical issue of this analysis consists of the accurate selection of all parameters as input for the numerical simulation and finite simulation which are often difficult under this circumstance. This work simplified the approach to compute mechanical stresses in rolling die of hot milling, based on a

plane strain subjected on its surface to mechanical loads for number of cyclic loads. The presented approach is obviously a preliminary investigation, in which the contribution is more methodological than quantitative. The model can be used as a tool in the most challenging aspects of hot rolling activities, concerning the identification of the elasticity damage mechanisms in rolling die. Actually, this approach looks quite promising, since it avoids analyzing the slab behavior in rolling gap, which is able to catch the relevant phenomena induced by rolling contact and identifies the radial and tangential stress behavior of the rolling die material.

ABAQUS offers many capabilities that enable stresses and failure modeling. Especially, Dynamic Explicit is designed for continuous and singularity-free problems, such as its application in damage evolution. A representative result of FEM analyses that is shown in Figure 15 is a progressive damage in which damage initiation and evolution are characterized by the accumulated inelastic stress energy per stabilized cycle. The capability uses a combination of Fourier series and time integration is fruitful to obtain the stabilized response of material structure. In short, the results agree with expectations and indicates the successful implementation of the constitutive model in ABAQUS.

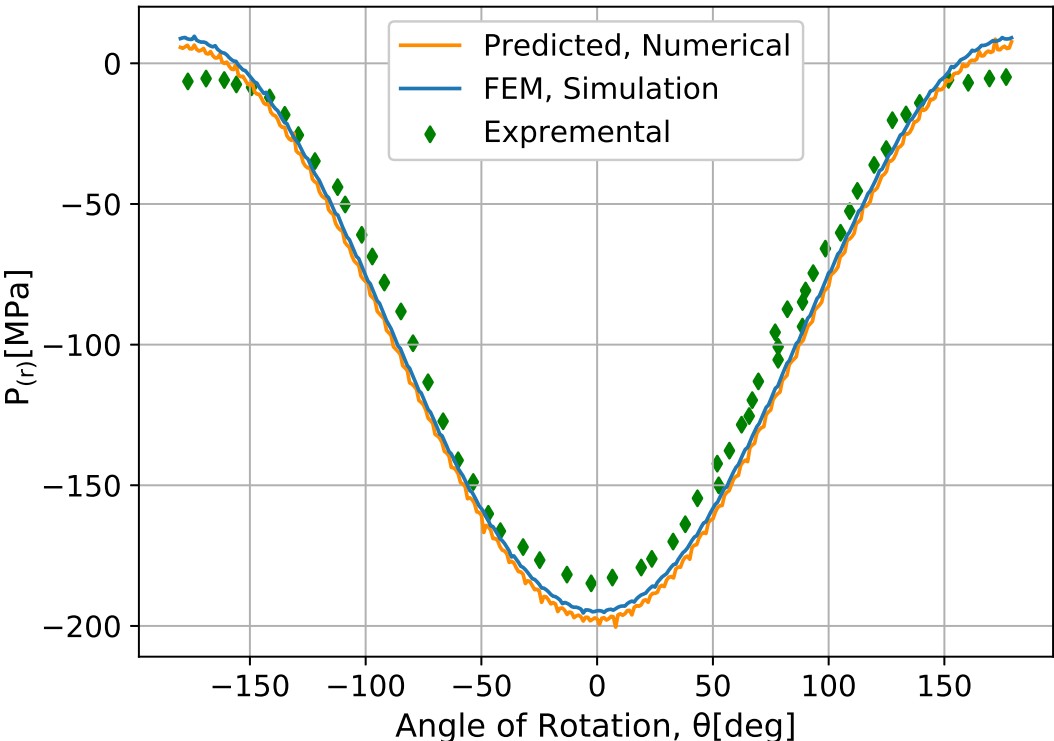

**Figure 14.** Radial stress distribution in rolling die under rolling gap for given number of cyclic loads (*n* = 1 × 10$^3$).

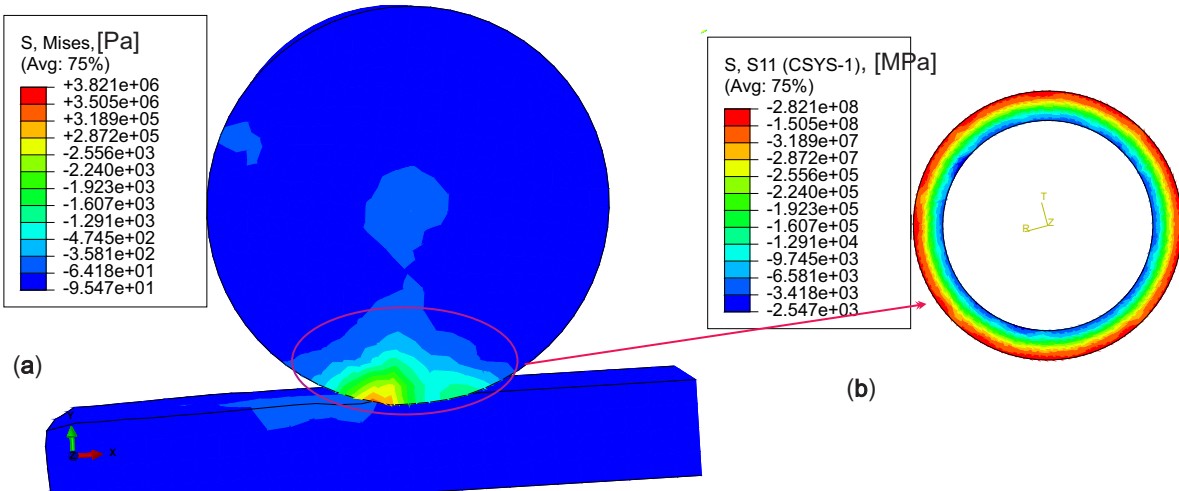

**Figure 15.** Rolling contact stress field under rolling gap results from ABAQUS. (**a**) von Mises stress distribution under rolling gap (Increment = $1 \times 10$), (**b**) Sub-model of radial stress distribution on the rolling die surface (Increment = $1 \times 10^4$).

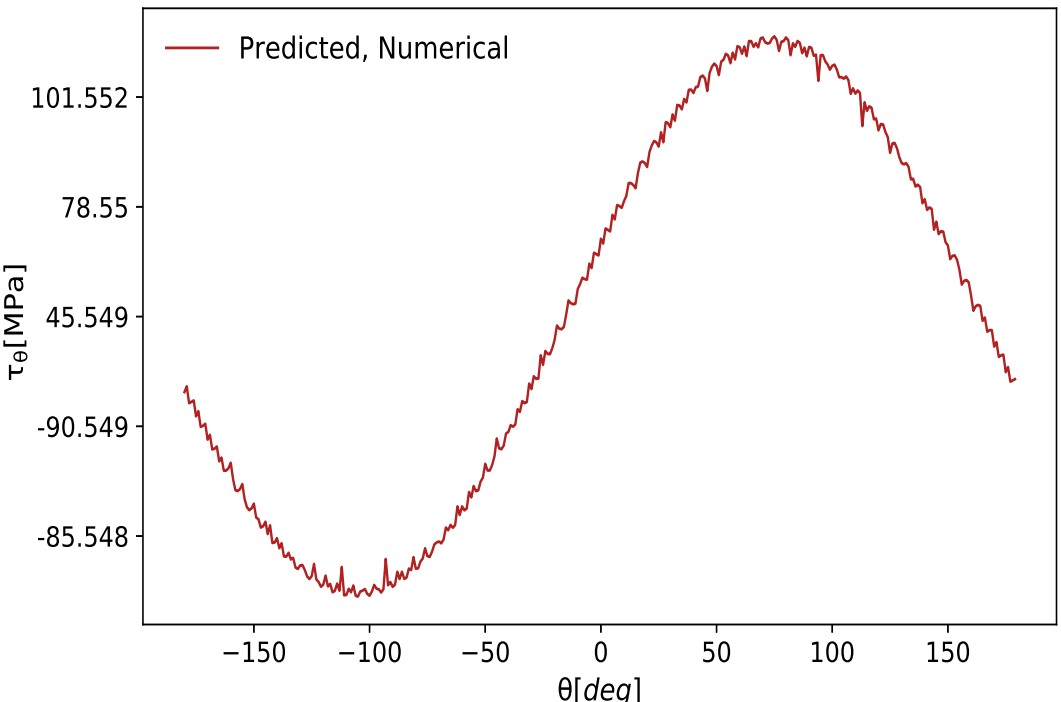

**Figure 16.** Circumferential stress distribution in the rolling gap for given number of cyclic loads.

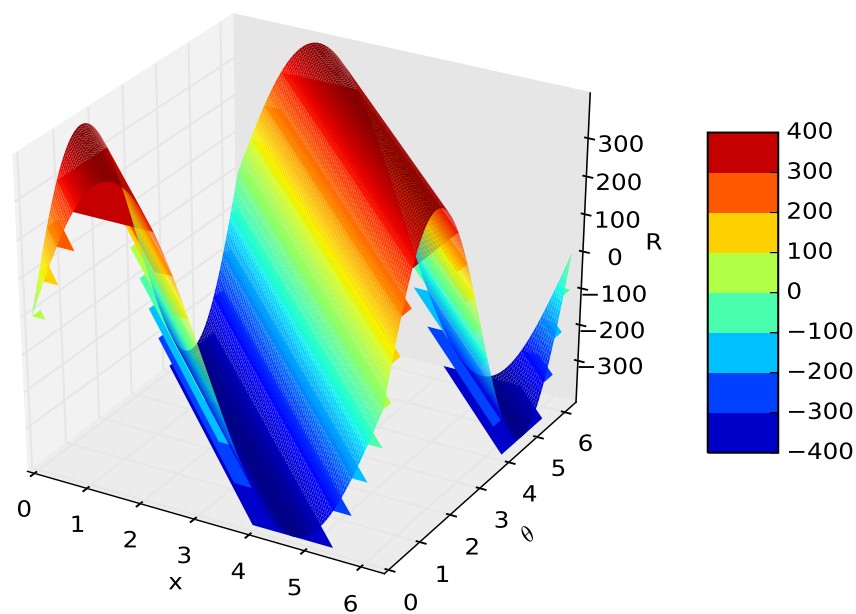

**Figure 17.** 3D surface folded under rolling contact due to compressive and tension loads in MPa.

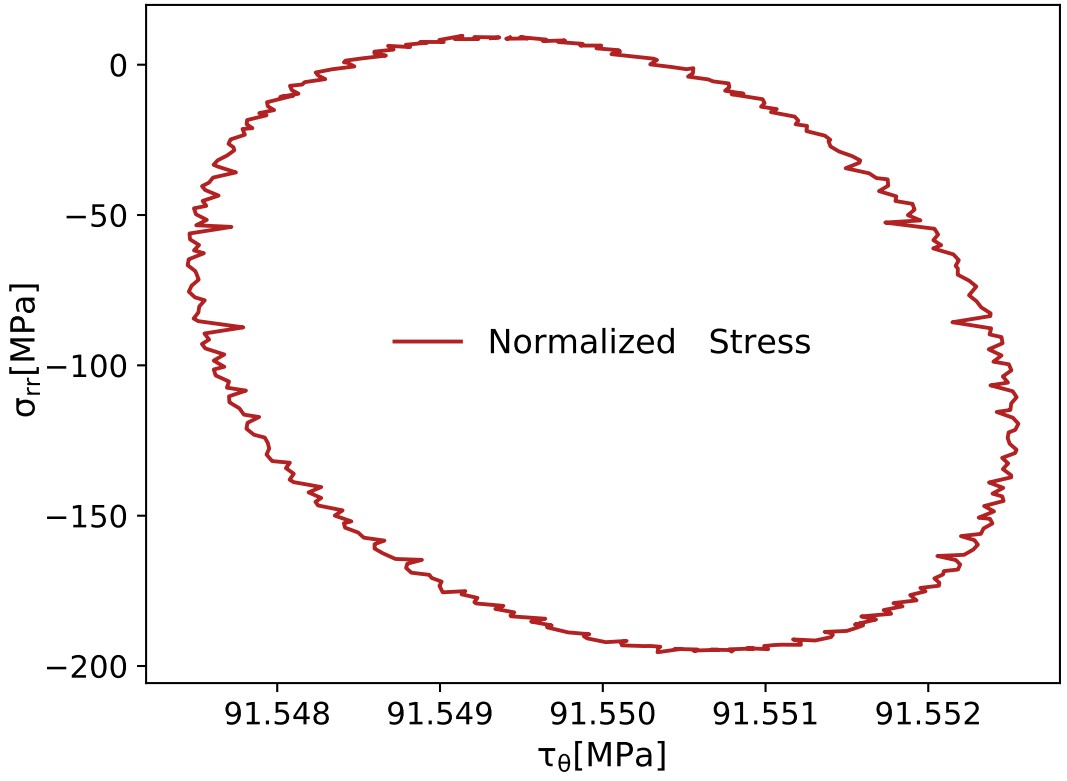

**Figure 18.** Normalized stress on the rolling die surface under rolling gap.

## 6. Conclusions and Outlook

The results from the numerical calculations are carried out based on a geometric relationship for rolling contact gap, normal contact and tangential stress distribution, neutral point, coefficient friction without including front and back tensions. This work proposes a simplified approach to compute radial and tangential stresses for the rolling die under hot milling for a number of cyclic loads/contacts. Stress distribution under rolling gap in a radial direction is computed by FEM simulations and numeric analysis, and the result shows good agreement with experimental data. All models look quite promising, since it avoids analyzing of the slab in the general solution and enables to catch the relevant phenomena which are induced by compressive and tension loads. Stress behavior on the rolling die is also identified with all possibility of the loads, which were poorly investigated in the literature. In fact, a deep experimental validation for the tangential coupled numeric model is strictly needed to allow a suitable model updating. However, the experimental setup is very complex, expensive and time-consuimg, which are difficult to be completed on existing hot milling, which is used daily in the manufacturing processes. Moreover, the work focuses on the modeling of hot milling elasticity damage evolution with full up to date information from the beginning, by assuming that all elasticity stresses can contribute to the damage evolution for a given number of cyclic contacts. Even though the results obtained are satisfactory for most of the models, there are limitations which can have negative effects on the utilization of the model or on its effective use for simulation. For instance:

- Complexity of boundary conditions and nature of application FEM implementation is difficult.
- Role of residual stress during manufacturing process on rolling die life is not considered in this work, etc.

For further investigation, thermo–mechanical stress and fatigue life prediction may be recommendable taking into account the results obtained in this study.

**Author Contributions:** M.A.T. did the former analysis and wrote the paper; H.A. was supervisor and participated in the editor.

**Funding:** This research has no received any funding from external and internal institutes.

**Conflicts of Interest:** The authors declare no conflict of interest.

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
