# Peer review of "Numerical Modeling Rolling Contact Problem and Elasticity Deformation of Rolling Die under Hot Milling"

_metals, doi:10.3390/met9020226_

Round 1
Reviewer 1 Report
The paper relates to an up-to-date issue of modelling contact problems in longitudinal rolling of strips. The authors investigate this problem by the slab method which is of rather historic importance and not in widespread use anymore. Nevertheless, I think that the paper offers a complex analysis of the problem and can be published in such journal as “Metals”. However, prior to publication, the following must be revised:
Due to the employed method, the authors refer in their study to classical works that were published a long time ago (2/3 of the cited references were published in the previous century). The Introduction section should therefore be extended to include up-to-date literature devoted to the problem, e.g. Wang et al. Chin. J. Mech. Eng. (2018) 31:46; Li et al. Metals 2018, 8(10), 783; Wang et al. Int J Adv Manuf Technol (2017) 92:1371–1389. There are really many recent publications on this problem.
The analysis is limited to the rolling cases wherein the shear stress on the contact surface is lower than the limit value equal to the yield point of material at pure shear. However, the authors do not give any applicability limits for the reported dependencies resulting from this condition.
The distribution of forces shown in Fig. 5 must be made more realistic: the friction force F is higher than the load, although F = μP, where μ<1.
Fig. 8 is unclear. What do authors mean by “variables”? If the variables are: temperature, velocity and reduction, then they should be indicated on the X-axis.
Fig. 10 shows two stress distribution patterns, however the figure does specify any difference between them.
It is necessary to add units of the parameters shown in Figs. 10, 15, 17.
Sometimes the authors give first names instead of family names of the cited authors, e.g. in 14, 21. The references cited in the paper must be revised in this respect.
Author Response
Many thanks for your comments. Regarding the response to the comments Pdf file is attached, let's find it.
With Regards,

Reviewer 2 Report
This paper entitled “Numerical Modeling Rolling Contact Problem and Elasticity Deformation of Rolling Die under Hot Milling", the authors models the rolling die contact with the slab primarily needs to describe the Tribology of contact. The topic is of interest to the Journal “Metals” readers to understand the model that includes new proposed constitutive equations for discontinuity of the velocity, pressure distribution in rolling contact from the entry side to exit side of the neutral point. However, some revisions need to be addressed in the present form of the paper to be considered for publication in this journal.
1 The introduction section of this paper is coarse and very short. It is recommended to add novel publications to describe the state of art of this topic like tempering and annealing processes in metals. Accordingly, please review the following recent articles and its references that have been study the thermomechanical coupling effects of rolling operation that affect the material’s microstructure: DOI: 10.1115/1.4037798 and Doi: 10.1016/j.matdes.2015.11.067. It can also help you to strengthen the discussion section because it is a thermomechanical problem.
2 It is necessary to describe in detail the rational of this manuscript. What is bringing new compared with similar research.
3 It is necessary to discuss Figure 12 and explain the differences between the three models. It will be great which are the differences between them.
4 At the conclusion section, I recommend to add a sentence to address which are the limitations of the present model.
Author Response
Many thanks for your comments. Regarding the response to the comments Pdf file is attached, let's find it.
Many Thanks,

Round 2
Reviewer 2 Report
The authors have greatly improved the manuscript, but the following points still need to be addressed:
-It is mandatory comment in the introduction or discuss sections that hot rolling inquires a thermomechanical coupling effects that can have a big influence in the stress and strain distribution. It is recommended to see Doi: 10.1016/j.matdes.2015.11.067.
- The last part of the introduction section (rational part of this work) does not properly describe which one is the contribution of this work respect to previous works.
-It its recommend to reorganize the conclusion in several bullet to properly address the main conclusions and limitations of this research work.
Author Response
Thanks for your comments for the second time. Regarding the response to your comment the file is attached, find it.
With Regards,

Round 3
Reviewer 2 Report
The article can be accepted despite that several English mistakes were found. If possible thoughrouly review the English along the manuscript.